# How Variation in Risk Allele Output and Gene Interactions Shape the Genetic Architecture of Schizophrenia

**DOI:** 10.3390/genes13061040

**Published:** 2022-06-10

**Authors:** Merve Kasap, Donard S. Dwyer

**Affiliations:** 1Department of Pharmacology, Vanderbilt University, Nashville, TN 37235, USA; merve.kasap@vanderbilt.edu; 2Departments of Psychiatry and Behavioral Medicine and Pharmacology, Toxicology and Neuroscience, LSU Health Shreveport, Shreveport, LA 71103, USA

**Keywords:** epistasis, gene–gene interactions, schizophrenia

## Abstract

Schizophrenia is a highly heritable polygenic psychiatric disorder. Characterization of its genetic architecture may lead to a better understanding of the overall burden of risk variants and how they determine susceptibility to disease. A major goal of this project is to develop a modeling approach to compare and quantify the relative effects of single nucleotide polymorphisms (SNPs), copy number variants (CNVs) and other factors. We derived a mathematical model for the various genetic contributions based on the probability of expressing a combination of risk variants at a frequency that matched disease prevalence. The model included estimated risk variant allele outputs (VAOs) adjusted for population allele frequency. We hypothesized that schizophrenia risk genes would be more interactive than random genes and we confirmed this relationship. Gene–gene interactions may cause network ripple effects that spread and amplify small individual effects of risk variants. The modeling revealed that the number of risk alleles required to achieve the threshold for susceptibility will be determined by the average functional locus output (FLO) associated with a risk allele, the risk allele frequency (RAF), the number of protective variants present and the extent of gene interactions within and between risk loci. The model can account for the quantitative impact of protective variants as well as CNVs on disease susceptibility. The fact that non-affected individuals must carry a non-trivial burden of risk alleles suggests that genetic susceptibility will inevitably reach the threshold for schizophrenia at a recurring frequency in the population.

## 1. Introduction

Schizophrenia is a devastating psychiatric illness that afflicts approximately 0.7–1% of the population with no known cure and mainly symptomatic treatment [1,2,3]. It shows a high degree of heritability, but the genetic liability is complex with polygenic origins [4,5]. Both common inherited gene variants of small genetic-effect sizes [6,7,8] and rarer de novo mutations with greater effects, such as copy number variants (CNVs) [9,10,11], contribute to the causation of this disease. Large genome-wide association studies (GWAS) in schizophrenia have identified over 100 candidate risk loci with several hundred associated genes [12,13]. The schizophrenia risk-gene candidates are highly enriched for essential and evolutionarily-conserved genes [14,15] and mutation-intolerant genes [16]. As the result of Hill–Robertson interference and balancing-selection mechanisms, this may lead to the persistence of risk variants in the genome despite liability for disease [15,16].

Based on an accumulation of evidence, genetic influences in schizophrenia appear to resemble gene specification of quantitative traits, such as height or skin color, rather than classical Mendelian patterns of inheritance [17]. Consequently, the genetic architecture of schizophrenia is multifaceted [18,19] and its major features, such as the number of risk alleles involved and their allele frequencies in the population, are still unfolding. Mathematical modeling has previously been used to characterize various aspects of complex human traits [20,21,22,23].

There are divergent views on the main features of the architecture of schizophrenia with some groups proposing that a limited number of robust causal variants are adequate to contribute to disease [24], whereas others suggest compound models of strong non-synonymous mutations and rare CNVs acting in concert with common background alleles [25]. More expansive models envision that susceptibility is determined by hundreds or even thousands of risk alleles with small individual effects [7,19,26]. A major goal of this research project is to develop a modeling approach for comparing and quantifying the relative magnitude of effect of single nucleotide polymorphisms (SNPs), CNVs and other mutations.

The role of gene–gene interactions or epistasis in the genetic liability for psychiatric disorders is likewise controversial. Although epistasis is acknowledged as a theoretical contributor, the magnitude of its effects is viewed as ranging from negligible [27,28] to highly significant [29,30,31,32]. Gene interactions may help to explain the so-called missing heritability in psychiatric disorders [33], and genetic connectivity is extensive among the risk genes for bipolar disorder [34] and major depressive disorder [35]. Davierwala et al. [36] reported that essential genes are much more interactive than non-essential genes and schizophrenia risk genes are highly enriched for ones that are essential for life and broadly conserved during evolution [15]. Consequently, we hypothesized that risk genes for schizophrenia are similarly more interactive than random genes, which may amplify the effect sizes of interacting variants [37]. If so, the findings would highlight the importance of being able to account for gene–gene interactions in models of genetic liability for heritable psychiatric disorders.

Some of the genetic variants associated with schizophrenia may actually be protective rather than detrimental [38], which complicates the assessment of the total risk burden in an individual. Therefore, we explore the possibility that protective variants as well as risk variants (under negative selection) could be identified by analyzing allele frequencies and odds ratio (OR) scores in relation to case versus control status. Moreover, the studies reported here seek to answer two pertinent questions that turn out to be related to one another: how many risk genes (including both risk and protective) contribute to threshold liability for schizophrenia, and what role do gene–gene interactions play in overall risk determination? As described here, we develop a novel mathematical model to address the first question, and then explore the importance of gene interactions in shaping the risk of schizophrenia.

## 2. Methods

### 2.1. Theoretical Background and Equation for Risk-Gene Quantification

According to polygenic threshold models [19], multiple hits at different risk genes are required to achieve threshold risk for schizophrenia. Certain combinations of risk genes will significantly elevate liability for disease.

To estimate the number of genes required to reach threshold, we developed key equations as described in Appendix A. Equation (1) describes the functional output from a risk-associated locus:(1)FLO=(RAF(x)+(1−RAF))
where x is the variant allele output (VAO or altered activity level of the gene affected by a polymorphism) and RAF is the risk allele frequency in the population. This term can be considered the average functional locus output (FLO) from two (or more) alleles at a genetic locus when one is affected by variation. It is important to add that a risk gene may harbor more than a single variant; several variants (e.g., SNPs) may collectively affect the expression/function of the gene. The FLO is intended to reflect the overall contributions of relevant variants of a particular risk gene. Thus, the total number of risk variants for a disorder that are present in the genome is different from, and greater than, the number of risk genes required to reach threshold for the disorder, which is the subject of this investigation.

Based on the probability of obtaining a series of variants with different VAOs and RAFs over multiple potential risk loci, we devised Equation (2):(2)$$P_{t}={\left[\mathrm{RAF}1\left(x1\right)+\left(1-\mathrm{RAF}1\right)\right]}^{y_{1}}\times {\left[\mathrm{RAF}2\left(x2\right)+\left(1-\mathrm{RAF}2\right)\right]}^{y_{2}}\times \cdots n$$
where $P_{t}$ is the probability of threshold risk-gene combination in the population, x and RAF correspond to above and y is the number of different risk genes associated with particular x and RAF values. We can substitute estimates of x and RAF and solve for y to calculate the number of risk genes needed to reach threshold for schizophrenia. We ran the model calculations until $P_{t}$ converged on 0.01, the prevalence of schizophrenia in the population.

### 2.2. Risk Allele Frequency (RAF) of Disease-Associated Variants

The PGC risk variant analysis included reference alleles that were both minor and major in terms of population expression; therefore, we referred to their frequency in the genome as RAF to reflect this. We characterized individual RAFs of the risk variants from the PGC dataset to derive representative estimates for plugging into Equation (2). To obtain OR scores from this dataset having the same directional relationship (>1), frequencies below 1 were expressed as 1/OR. From the original list of 128 variants, we excluded 14 indels from our analysis. For comparison, we generated a random list of genes with Molbiotools’ Random Gene Set Generator (Available online: http://www.molbiotools.com/randomsequencegenerator.html accessed on 11 May 2020). We then randomly selected SNP variants from these genes to compile a similar-sized list of reference variants. The population allele frequencies of the randomly chosen reference variants were recorded.

In addition, we characterized the allele frequencies of a subset of risk variants that were associated with syntenic blocks of functionally related genes, as previously described [15]. The syntenic blocks examined here corresponded to the final ranking of variants assigned in the PGC study [12]: 9, 22, 31, 38, 40, 47, 59, 84 and 85. We determined the RAFs and OR scores for this subset of alleles and compared these values with those from the complete list of risk variants.

### 2.3. Gene-Interaction Analysis

Gene interactions are defined here as pairs of genes that, when co-retained in cells, affect cell survival, as determined by Lin et al. [39], with radiation hybrids. These genes and networks comprise the “Genetic Interactions” data used by GeneMANIA [40].

To compare genetic interactions in the entire PGC gene dataset with interactions between genes randomly selected from the human genome, we used GeneMANIA and selected “Genetic interactions” with “Max resultant genes” (other genes) set to 0. For comparison, we generated 4 non-overlapping random gene lists (as above) of roughly the same size (333–346 genes) as the final PGC dataset (338) and analyzed these under the same conditions. In total, 1350 random genes were analyzed and inclusion of additional genes would not appreciably change the outcome. We also compared a rheumatoid arthritis (RA) dataset of 326 genes derived by Okada et al. [41].

We quantified the genetic interactions as previously described [34]. We then used the number of links per gene to compare the connections involving the set of PGC risk genes with those of the randomly selected gene lists.

### 2.4. Statistical Methods

We compared the RAFs of the PGC SNP dataset to a set of randomly generated SNPs using a standard *t*-test.

To determine the statistical significance for comparisons between the number of genetic interactions among genes in the different sets, we first calculated the mean and standard deviation from data derived from 4 random lists of genes expressed as the number of links per gene obtained from GeneMANIA, as previously published [34,35]. Based on the standard deviations from the random gene lists, we established confidence intervals (CIs) for significant differences at the *p* < 0.01 or *p* < 0.05 levels. We then determined whether the data from the PGC dataset surpassed the CIs calculated from the four random gene sets.

## 3. Results

### 3.1. Distribution of RAFs

We sought to determine if there was evidence of negative and positive selections on the PGC risk variants and to derive an average allele frequency for entering into Equation (2). Therefore, we characterized the allele frequencies of risk variants associated with schizophrenia and obtained profiles of cases vs. controls (Figure 1) to obtain estimates of the average RAF. We compared RAFs from the PGC dataset to the population allele frequencies of 114 randomly generated SNPs and discovered no difference between the allele frequencies for these two groups of SNPs—both averaged 0.44. Thus, schizophrenia risk alleles showed the same overall frequency profile as a random set of alleles, which is similar to the data of Ohi et al. [42] on an overlapping gene set (average RAF 0.46). This argues against a strong selection of the risk variants, consistent with the findings of Liu et al. [43] and Yao et al. [44], and against these common variants belonging to a subset with unique characteristics.

A closer examination of the highest and lowest quartiles revealed additional interesting relationships. Regression analysis showed a stronger positive correlation between RAF and OR score for the highest quartile of variants occurring more often in controls than cases (Figure 2), which suggests that they might represent protective variants. A opposite relationship was observed for the lowest quartile, perhaps indicating that adverse risk variants are under stronger negative selection. Positive and negative selections have both been previously observed for variants associated with susceptibility to schizophrenia. Overall, the data support the findings of Nishino et al. [38] that protective variants co-occur in the genomes of individuals with schizophrenia and should be accounted for when estimating the burden of risk due to genetic factors.

In a previous study, we identified syntenic blocks of risk-gene candidates in schizophrenia that were collectively geared toward common purposes [15]. When we characterized the RAFs and OR scores for these 9 blocks, we found that the SNP variants did not differ significantly from the total SNP set in terms of RAF (0.47 ± 0.16 vs. 0.44 ± 0.26) or OR score (1.075 ± 0.009 vs. 1.086 ± 0.036), and they fell into the middle two quartiles of the overall data (Figure 1A). This distribution suggests that the alleles associated with syntenic blocks of risk genes are not under strong selection. The lack of obvious selection on these functional blocks of genes suggests that they may reside in genomic regions, such as recombination cold spots or regions that experience background selection with reduced mutability in order to preserve the local adaptive arrangement.

### 3.2. Risk-Gene Quantification

To assess the underlying risk-gene architecture, we used Equation (2). If the risk variants decrease the output from the affected alleles to 80% of normal, $P_{t}$ will reach the 1% criterion at 50 genes with an average RAF of 0.44 (Figure 3A). By contrast, if the VAO is 95% of the non-risk allele and the average frequency of expression is low (RAF = 0.1), then it would require 900 risk alleles to reach the 0.01 threshold (Figure 3B). Thus, the number of risk alleles needed to achieve the threshold for disease decreases as the frequency of the risk allele in the population increases and in relation to the relative loss of functional output from the affected locus. This relationship is similar to the model calculations of Wray and Visscher [19], and demonstrates the utility of simple models as starting points for analyzing the genetic architecture of schizophrenia.

### 3.3. Inclusion of Protective Variants

The work of Nishino et al. [38] and logical considerations converge on the conclusion that individuals with schizophrenia will harbor some protective variants in their genome. In addition, Hess et al. [45] recently identified resilience variants that modify the risk for schizophrenia. In comparison to protective variants, which are typically the alternate allele at a risk locus, resilience alleles moderate the adverse effects of risk variants. To account for the moderating effects of both types of protective alleles, we re-examined the model with inclusion of values for x > 1 to reflect the protective variants in Equation (2). For simplicity, we chose a value for the FLO of 1.015 for the simulations, because it is equal in magnitude but opposite in effect of the FLO used for adverse risk variants (0.985). Protective variants will actually have an array of effect sizes similar to the risk variants. Figure 3C reveals that the number of risk variants required to reach threshold increases as a function of the number of protective variants included in these simulations. Moreover, this modification of Equation (2) allows us to quantitatively model the effects of protective variants on overall disease risk.

### 3.4. Genetic Interactions among Risk Variants

Based on the previous investigation of risk genes for bipolar disorder [34], we hypothesized that the PGC risk genes will also show a greater interaction with each other than would be observed in randomly selected sets of genes. Gene interactions can potentially amplify the effects of risk variants and we refer to this phenomenon as network ripple effects.

The results of GeneMANIA analysis are depicted in Figure 4, Figure A1 and Figure A2. For the PGC risk-gene set, we detected a significant increase in the number of interactions per gene among members of this list compared to the random genes (Figure 4). Gene interactions provide information about the network connectivity of risk genes and appear to be correlated with the degree of evolutionary conservation of the genes. The latter observation may stem from a more extensive integration of older genes into functional networks. Therefore, gene interactions may be an important source of added liability in schizophrenia and may explain some of the missing heritability.

### 3.5. Effect of Gene Interactions and CNVs on Models of Genetic Risk

The extensive interactions among schizophrenia risk genes forces us to consider such interactions when estimating the number of risk alleles that predispose to disease. At a single SNP locus, multiple nearby risk-gene candidates may be present, which functionally interact either locally or with genes on different chromosomes. If the average VAO due to genetic variation is 0.965, it will result in a FLO of 0.985 when adjusted for average RAF (0.44). If this is a single independent site, then 0.985 will go into Equation (2). Likewise, the value of 0.985 would represent the contributions of a second independent locus with a similar output. However, when two risk genes interact, it is essentially a non-random assorting of that gene combination. To mathematically represent these non-random or non-independent combinations, we chose to assign a value of $0.985^{2}$ (0.97) instead of 0.985 to each interacting locus. If a risk gene interacts with more than one additional risk gene, then for calculation purposes, the interacting gene FLO would be $0.985^{2+n}$, where *n* in the exponent equals the number of additional interacting risk genes.

To model the gene interactions, we used 30% as the fraction of interacting risk genes, which is close to the number derived from our previous work [15]. As can be observed in Figure 5A, ~300 risk alleles are required to achieve the 0.01 threshold in the case of no genetic interactions, whereas this number falls as the interactions increase. Providing this is a useful way to quantitate gene interactions; this means that gene–gene interactions enhance the liability of the small-effect sizes of individual risk variants.

CNVs represent an additional source of compound genetic interactions because they can potentially affect the expression of functional blocks of genes. Therefore, we sought to model the estimates of the number of risk genes needed to achieve threshold risk for schizophrenia in the absence or presence of CNV FLOs (0.5) in Equation (2). A single CNV replaces about 15% of the total risk genes needed to achieve threshold, whereas two CNVs reduce that number by 30% (Figure 5B). Therefore, CNVs make significant contributions to disease liability; however, their effects must occur on a substantial background of risk variation. For example, in Figure 5B, a single CNV would replace the contributions of approximately 45 common risk alleles, but not obviate the need for many additional ones (250–260 in this case).

Taken together, we speculate that the total number of genes (risk + protective) that will achieve threshold for schizophrenia is in the range of 2500–2600. This estimate is based on solving Equation (2) with the following parameters: an average VAO of 0.995 (a 0.5% reduction in gene expression/output due to the polymorphism), an average RAF of 0.44, which results in a FLO of 0.9978, 30% of the risk genes interact with one additional partner and 25% of the variation affects protective genes (FLO = 1.0022). This “best guess” estimate of ~2500 risk **genes** is closer to the assessment of 8300 causal **variants** (note: each gene may have multiple variants associated with it) by Ripke et al. [46] than to the 40,000 causal variants suggested by Nishino et al. [38]. By contrast, Zhang et al. [23] suggested more than 10,000 susceptibility SNPs may be involved in schizophrenia based on the same gene dataset, whereas Frei et al. [47] estimated 8500 variants that explained 90% SNP heritability, which agrees with the assessment of Ripke et al. [46] and our calculations. Holland et al. [21] used mathematical modeling to arrive at 31,000 causal SNPs. Some of the tendency to overestimate the number of contributing risk genes may be due to the differences in what is considered a risk variant. Moreover, the failure to incorporate gene–gene interactions in other estimates ignores the possibility that genetic interactions may amplify the effects of individual variants and decrease the genetic burden needed to reach threshold for the disorder.

## 4. Discussion

The major novel findings of this study include: (1) a simple equation relating altered genetic output from variant alleles and RAFs to the total number of risk genes required to reach threshold for schizophrenia, (2) support for the occurrence of protective variants undergoing positive selection and a means to deal with them computationally and (3) the discovery of extensive gene–gene interactions among schizophrenia risk genes and a strategy for including them in genetic liability calculations. Finally, we provided a quantitative basis for harmonizing views about the relative contributions of SNPs and CNVs to the genetic architecture.

Wray and Visscher [19] made impressive headway in characterizing the genetic architecture of schizophrenia and our analysis generally complements theirs. Specifically, we developed a straightforward mathematical model for estimating risk-gene burden with explicit terms for RAF, the effect size due to risk-gene variation (VAO), gene–gene interactions and inclusion of protective variants. The model provides a snapshot of the different factors that influence the genetic liability for schizophrenia. Parameters can be varied to take into account as much complexity as desired.

Analysis of the PGC data revealed that there was an equal number of schizophrenia risk variants with increased vs. decreased occurrence (negative vs. protective effects?) in the case population. Previous studies have revealed evidence for both positive and negative selection of variants that show differences in frequency comparing control subjects with those affected with schizophrenia [16,43,44,48]. For an optimized phenotype, most mutations are expected to produce a negative effect because there is a statistically greater chance of an adverse outcome the closer the phenotype is to optimum. Since this trend was not observed in the PGC dataset, it suggests that most of the traits underlying schizophrenia are not optimized at this point or perhaps are not optimizable due to pleiotropy.

The work of Nishino et al. [38] and Hess et al. [45] suggest that many protective variants will be expressed in the genomes of the case population. These protective (resilience) variants counterbalance the adverse effects of risk variants [45]. Therefore, the balance between adverse and protective alleles must tilt substantially toward the former to produce symptoms of schizophrenia. Taken together with the observation that risk variants for schizophrenia are enriched in essential genes [15], these findings suggest that the negative selection of adverse risk variants is minimal, perhaps because the associated genes have been optimized for critical pleiotropic purposes. Furthermore, SNPs associated with schizophrenia risk may affect the expression of multiple genes, sometimes with opposing functions [49]. Therefore, the net result on relevant behavior may reflect the sum of effects on many genes [49], which can be accounted for with our concept of functional locus output (FLO).

A special subset of variants associated with syntenic blocks of genes exhibited intermediate RAFs and average OR scores under little apparent selective pressure. We previously speculated that the process of gene amalgamation to form these syntenic blocks may have been accompanied by creation of recombination cold spots, which would impede selection but also preserve weak-risk variants in the DNA blocks [15]. The data presented here support this earlier suggestion.

Consistent with the studies of gene interactions in bipolar disorder and depression [34,35], candidate risk genes for schizophrenia showed greater network interaction than random gene sets of the same size. These gene–gene interactions may cause network ripple effects among the connected genes that amplify the impact of small individual VAOs. Furthermore, this notion is similar to that of Greenspan [50] who described the functional connectivity of gene networks in terms of flexibility and pleiotropy. Despite some overlap with the omnigenic model of Boyle at al. [26], there are also important distinctions. At some level, all genes are interconnected based on how the genome evolved [51]. This common origin can obscure real differences between genes and disorders. For example, the risk genes for bipolar disorder and depression are much more interconnected (2–3-fold-higher number of gene interactions) than the schizophrenia risk genes. If the various risk genes participated in omnigenic interactions, we would not expect to observe these considerable differences, especially in psychiatric disorders with overlapping symptoms and involving similar cell types.

Our studies emphasized the importance of gene–gene interactions (epistasis) in modeling the overall genetic architecture. By contrast, statistical approaches to epistasis concluded that genetic interactions make minimal contributions to variance in complex traits and disorders [52]. However, epistasis is clearly a significant genetic process as amply demonstrated in model organisms, such as *Caenorhabditis elegans* and *Drosophila* [53,54]. The dichotomy between statistical epistasis and compositional/functional epistasis has been previously noted [31,53]. The effect of a genetic variant (defined here as VAO) occurs in the context of the broader genetic background, which means gene interactions must be accounted for in the model. Furthermore, the pathways, proteins and genes involved in schizophrenia are organized into tangible networks [55,56,57] that mediate the effects of genetic variation. Here, we attempted to quantitatively assess the impact of gene interactions on overall risk burden. Finally, the characterization of the gene interaction networks may identify key hub genes that could represent future targets of therapeutic intervention.

Multiple factors contribute to the total risk burden for schizophrenia: risk variants (SNPs, CNVs), protective variants, gene interactions, allele frequency and the overall effect size of the genetic mutation. We represented these various factors in a simple equation that can reconcile relative contributions from common non-coding SNPs and CNVs. CNVs, null mutants and major functional mutations (loss or gain of function) will produce significant liability for schizophrenia; however, their effects manifest on a background of significant risk alleles [58,59,60]. Bergen et al. [60] reported that individuals diagnosed with schizophrenia who showed that copy number variation in risk genes had lower polygenic risk scores (fewer small-effect SNPs) than patients without CNVs. Similarly, Zhou et al. [61] observed that Tourette Syndrome patients with large-effect variants (LEVs) or CNVs required fewer small-effect common variants to reach threshold for disease than TS patients who lacked large-effect variants. These clinical findings provide support for the model presented here. Using the estimated parameters, we calculated that a deletion CNV was roughly equivalent to around 15% of the total single-nucleotide variants required to reach threshold for disease. By itself, a single CNV in a risk gene would not be sufficient to cause schizophrenia without the contributions of many additional background risk variants.

Interpretation of the genetic architecture in schizophrenia must be considered cautiously due to various inherent limitations. GWAS and CNV studies have likely identified instances of false-positive candidate genes, whereas some actual risk genes may have been missed. Rare MAFs have thus far been largely neglected; however, these rare alleles are likely to have large effects based on previous work [7,60,62]. Environmental factors and epigenetic changes also complicate the interpretation of genetic influences on disease susceptibility. We limited our simulations to average values for parameters, such as RAFs and VAO, so the real situation is much more complicated; however, the equation can manage greater complexity than presented here. Nevertheless, overall trends, such as a requirement for a substantial number of risk variants to reach threshold and the potential significance of genetic interactions, are based on solid observations and logic. Finally, there may be alternative ways to handle gene–gene interactions mathematically; however, this work provides a useful conceptual framework for the problem.

According to the model, non-affected individuals must harbor a non-trivial complement of risk alleles that experience little selection. Furthermore, threshold combinations of risk alleles will be inherited at a set frequency in the population—a phenomenon previously described as inevitable bad luck [15]. Therefore, schizophrenia differs significantly from certain pathological genetic conditions, such as inherited metabolic disorders or rare Mendelian diseases. Instead, schizophrenia risk variants via their different VAOs and RAFs determine whether certain quantitative traits fall in the normal range. In our distant ancestors, the concurrent expression of various suboptimum traits may have carried little penalty for the individual. Without the complexity, artificiality and stress of modern society, someone showing a collection of traits that would be diagnosed today as schizophrenia may still have been largely functional when living a simpler existence in nature. This emerging view of schizophrenia has important implications for diagnosis and treatment. In terms of diagnosis, the genetic inevitability and diversity of schizophrenia calls into question the value of a monolithic diagnostic category and argues instead for the assessment of symptoms according to quantitative trait dimensions along the lines of the Research Domain Criteria (RDoC) [63]. If schizophrenia is best conceived as a collection of suboptimum traits, then treatment requires a multimodal approach with antipsychotic medications, other psychotropic drugs, behavioral therapy and cognitive enhancement all part of the plan. The complex genetics of schizophrenia dictate a correspondingly complex therapeutic strategy.

## Figures and Tables

**Figure 1 genes-13-01040-f001:**
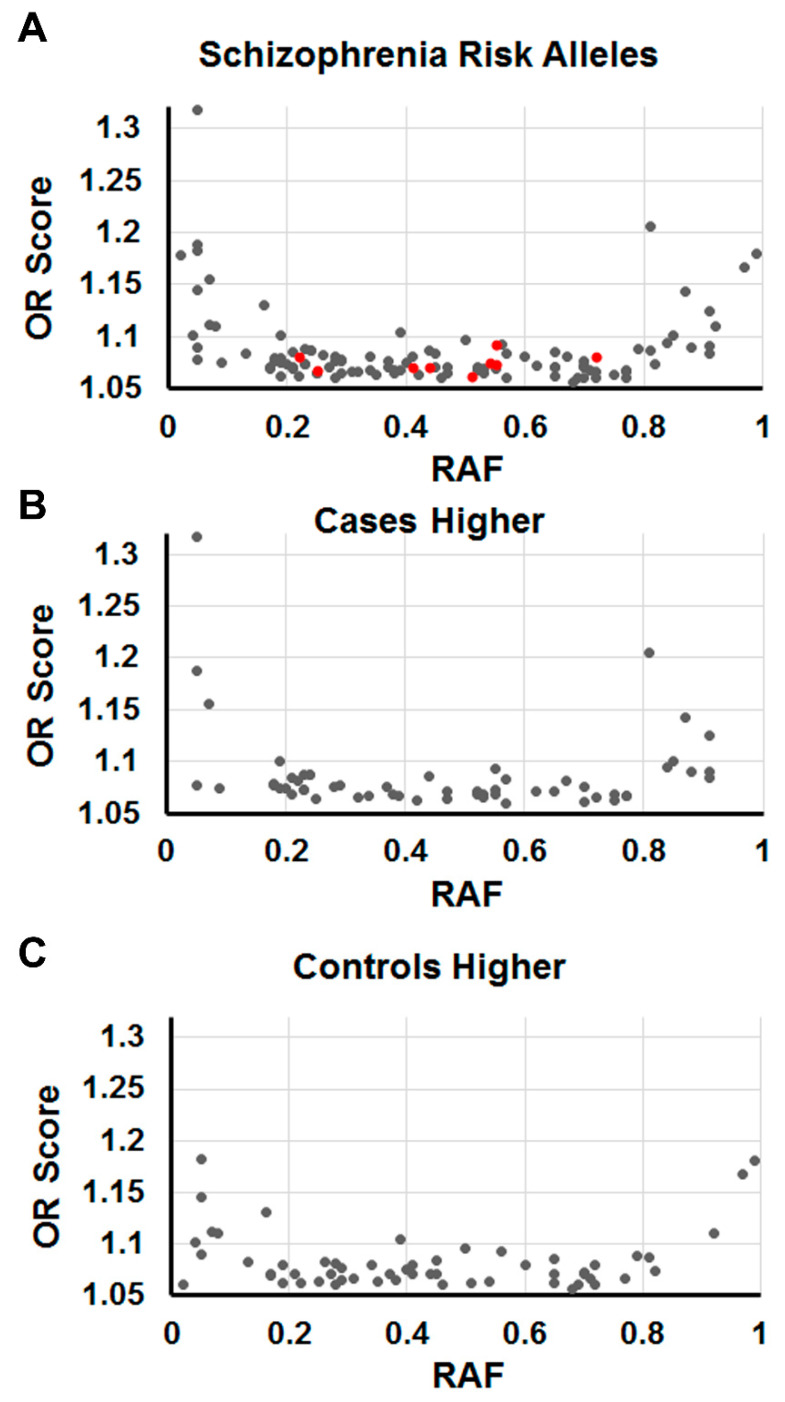
Distribution of OR scores as a function of RAF for the PGC risk alleles. (**A**) Data from 114 risk alleles are plotted. The red circles represent SNPs associated with functional syntenic blocks as described in the text. (**B**) Plots of 57 SNPs that occurred more frequently in cases of schizophrenia than controls, and (**C**) 57 SNPs that were higher in controls than cases. It is worth noting that the case and control distributions are superficially very similar to each other.

**Figure 2 genes-13-01040-f002:**
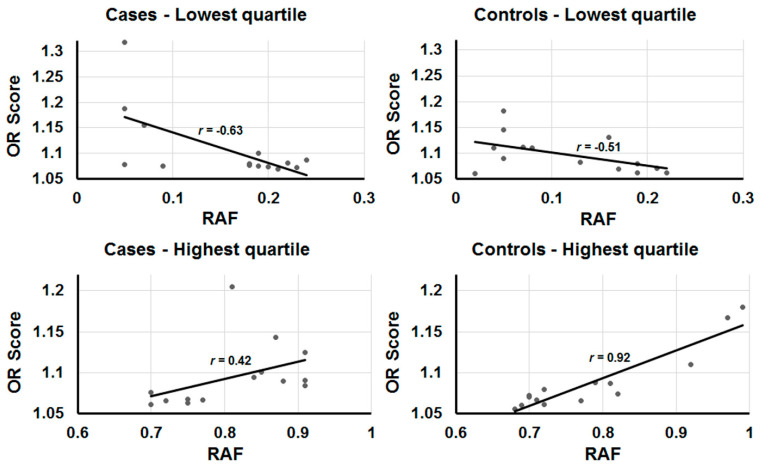
Regression analyses comparing RAF with OR score. Data from the highest and lowest quartiles were evaluated by standard regression analysis for relationships between their allele frequencies and adjusted OR scores. The plots show the lowest and highest quartiles of data points from SNPs that occur more frequently in cases than controls and vice versa (as labeled). The trend lines are included along with the correlation coefficient, *r*.

**Figure 3 genes-13-01040-f003:**
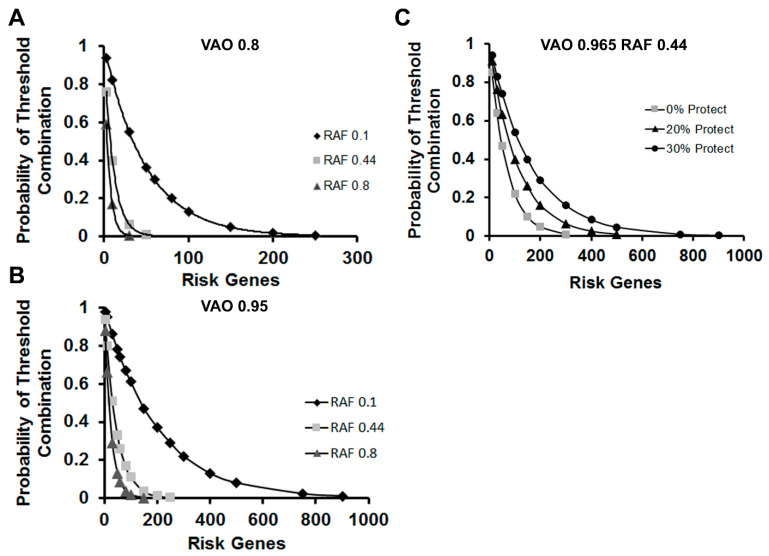
Risk-gene estimation. Graphs were rendered for the number of risk genes required to reach the 0.01 probability of threshold risk-gene combination in the population (*P_t_*) from pairs of alleles associated with risk polymorphisms. The VAOs are labeled as 0.8 (**A**) and 0.95 (**B**). The average RAFs used in Equation (2) are indicated in the inset legends. (**C**) Estimation of risk alleles needed to reach threshold with protective variants included either 20% or 30% of the total (as labeled). We assumed that the protective variants had a VAO of 1.035 with a resulting FLO of 1.015, which was equal in magnitude, but opposite in effect from the VAO of 0.965 (FLO = 0.985) for adverse risk variants.

**Figure 4 genes-13-01040-f004:**
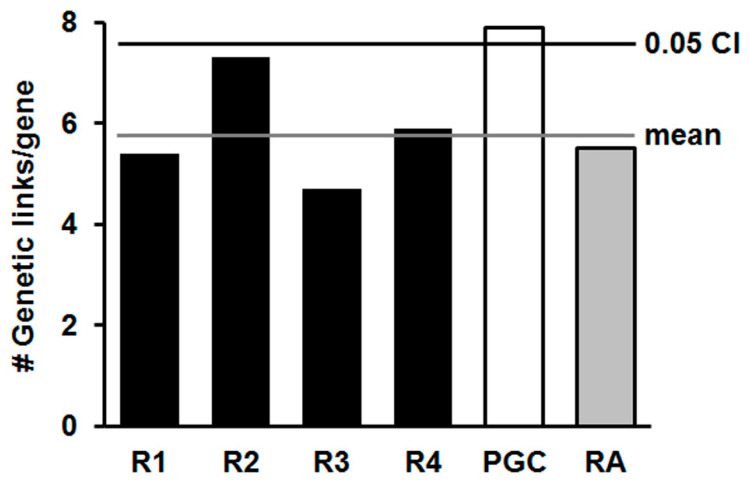
Genetic interactions among PGC risk genes. (A) The degree of genetic interaction among genes from 4 random gene lists (R1-R4) was analyzed with GeneMANIA and compared with the degree of interaction among risk-gene candidates from the PGC dataset and a rheumatoid arthritis (RA) dataset. The data are plotted as the number of genetic links per gene in the input lists that show interactions in the Lin et al. [39] dataset. The average of the 4 random lists is indicated with a gray line and the CI (*p* < 0.05) with a black line.

**Figure 5 genes-13-01040-f005:**
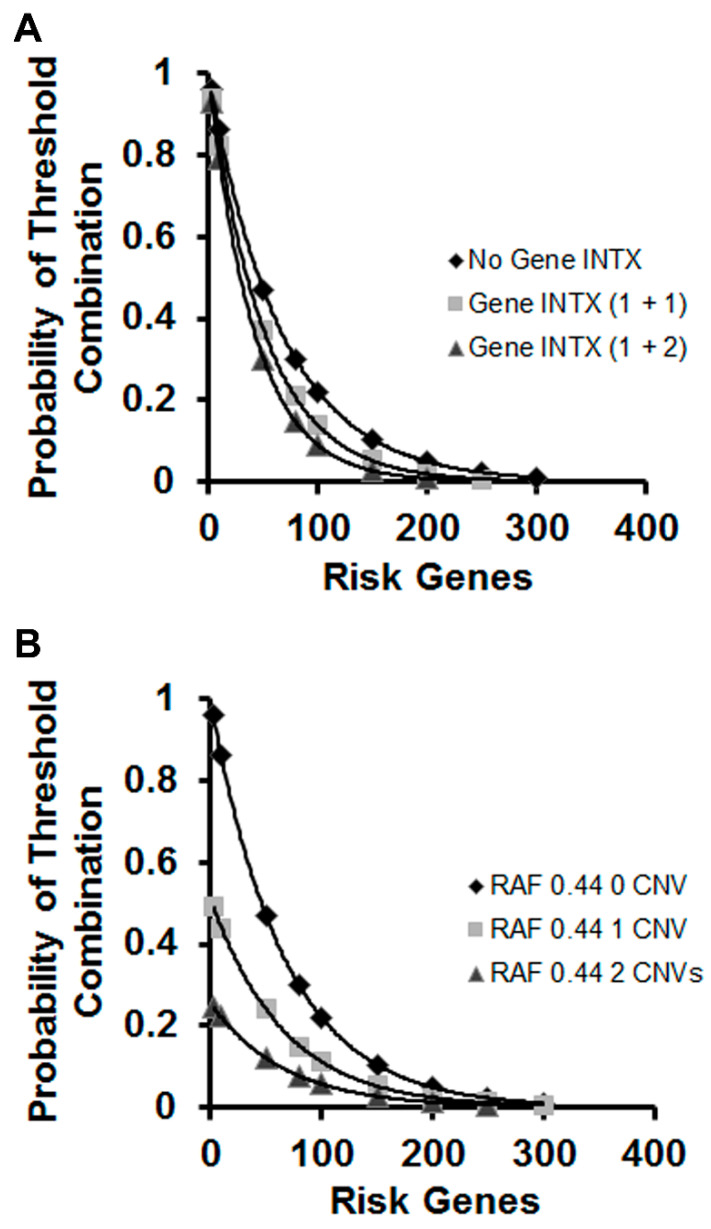
Estimation of risk-gene threshold with inclusion of gene–gene interactions and CNVs. (**A**) Data are plotted for the number of risk genes required to reach the 0.01 probability of threshold combination when gene–gene interactions are included in the calculation. For this model, the VAO is 0.965 and the FLO 0.985 with RAF is set at 0.44. Of the total risk genes, 30% are modeled as interacting with 1 (1 + 1) or 2 (1 + 2) other genes, which require substitution of either 0.97 (0.985^2^) or 0.956 (0.985^3^) into Equation (2) for genes with 1 or 2 partners, respectively. For comparison, calculations with no gene interactions (INTX) are also shown. (**B**) With the same settings as (A), the model includes 0, 1 or 2 CNVs, which are represented with FLOs of 0.5.

## Data Availability

All data have been included in the paper.

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
