# Peer review of "How Variation in Risk Allele Output and Gene Interactions Shape the Genetic Architecture of Schizophrenia"

_genes, 2022, doi:10.3390/genes13061040_

Round 1

Reviewer 1 Report

Kasap et al. used algorithmic modeling and revealed the highly interactive nature of schizophrenic risk genes. While suggesting some experts in genetic algorism to serve the peer reviewer’s role better, my comments are:

1. It’s not clear the contribution of this study to the diagnosis and treatment of schizophrenia.

2. Can you find the SNP data from a published case study of a schizophrenia patient with one or two CNV mutations to support your modeling in Figure 5B?

3. The images in Appendix B Figures 1 and 2 are unclear. Some of the gene names are hard to read. Please change the fonts.

Author Response

  1. It’s not clear the contribution of this study to the diagnosis and treatment of schizophrenia.

We thank the reviewer for this comment. Although this work is mainly aimed at characterizing the genetic architecture of schizophrenia, it is relevant to diagnosis and treatment. To point out this relevance, we have added text in the Discussion on pgs. 11-12.

  1. Can you find the SNP data from a published case study of a schizophrenia patient with one or two CNV mutations to support your modeling in Figure 5B?

This is also a good suggestion. At least two published studies have shown the inverse relationship between the presence of CNVs in actual patients and the polygenic risk score or total SNP liability. These papers have been quoted on p. 11. These clinical findings support the conclusions of our mathematical modeling. We have also added pertinent references to the list of cited publications.

  1. The images in Appendix B Figures 1 and 2 are unclear. Some of the gene names are hard to read. Please change the fonts.

We used the largest font available in the GeneMANIA program, which has limited options for customization. The main intent of these two figures was to show the general level or density of connections for a set of random genes in comparison to schizophrenia risk genes. There is also a tradeoff between capturing the entire network of connections and the size of individual gene icons. We have added text to this effect in the figure legend to Appendix B Figure 1. The zoom function on my computer also allows me to make out the gene names, if desired.

Reviewer 2 Report

Drs. Kasap and Dwyer reported a very interesting study elaborating the possibility that protective variants as well as risk variants could be identified by analyzing allele frequencies and OR scores in relation to case versus control status in schizophrenia. They tried to address two important questions, how many risk genes contributing to threshold liability for schizophrenia and what role do gene interactions play in overall risk determination. The manuscript was very well-presented. The study design was impressive, and results were well documented. Statistical analysis was thorough and reliable. This study is clearly relevant to the field.

Author Response

No issues to address. We thank the reviewer for appreciating the significance of the paper and for their complimentary comments.